# Emergency Management against Natural Hazards in the Acropolis of Athens

**Miranda Dandoulaki** [1], **Ioannis Evripiotis** [2], **Maria Gaspari** [3], **Miltiadis Katsaros** [2], **Eleni Linaki** [2] **and Konstantinos Serraos** [2],*

1 Disaster Prevention Research Institute, Kyoto University, Kyoto 611-0011, Japan
2 School of Architecture, Urban Planning Research Laboratory, National Technical University of Athens, 10682 Athens, Greece
3 World Bank, Global Facility for Disaster Reduction and Recovery, London NW33 JG, UK
* Correspondence: kserr@central.ntua.gr

**Abstract:** Using the case of the Acropolis of Athens, this paper aims to broaden current knowledge on risk and emergency management in archaeological complexes of high visitation. More specifically, it focuses on the protection of visitors and staff and intends to provide guidelines towards an emergency response plan for geodynamic and meteorological hazards in the Athens Acropolis archaeological site, along with a risk reduction and preparedness strategy. To this end, the paper first analyzes the main challenges arising from the everyday use of the archaeological site and the high visitor flows, mainly during summer. Secondly, it sets out the main parameters for drawing up an emergency evacuation plan for staff and visitors. Finally, it proposes preparedness guidelines, including training and information for all involved, together with a roadmap towards reducing existing risk and the implementation of necessary infrastructure interventions against residual risk. To finish, we conclude that challenges in emergency planning for the Acropolis of Athens do not arise solely from the unique conditions of the place and restrictions associated with the universal value of the monument but point towards reducing high residual risk and improving risk governance.

**Keywords:** evacuation planning; exposure; governance; archeological site; risk

## 1. Introduction

The Acropolis of Athens, settled on a 156-meter-high rocky hill in the center of the Athenian agglomeration, constitutes the main historic, archaeological, symbolic, and touristic asset of the capital city, Athens [1]. An outstanding symbol of the golden century of Pericles (5th century BC), the highest cultural acme of antiquity, the Acropolis was the main sanctuary of the ancient city, dedicated to its patron goddess, Athena. In 1987, it was declared a UNESCO "World Heritage Monument" [2], a fact that set it as one of the most important archaeological sites worldwide (Figure 1).

Due to its distinguished tangible and intangible cultural value, the Athens Acropolis is one of the most important touristic points worldwide, attracting throughout the year large numbers of visitors. At the same time, it is exposed to significant hazards of geodynamic and meteorological origin [3–7], and global climate change has already had observable effects on the natural, urban, and cultural environments of the region (Attica) [8]. As a result, managing risk and emergency planning have become priorities among the other challenges that the Acropolis faces.

Disaster risk reduction against natural hazards and the safeguarding of cultural heritage in case of an emergency are widely recognized as priorities for protecting cultural heritage [9–13]. Nonetheless, research addressing the protection of visitors and staff and not of the monument itself is limited [14,15]. Subsequently, our research can be considered as an initial step toward a more comprehensive and multi-disciplinary approach to risk and emergency management in archeological sites of high visitation.

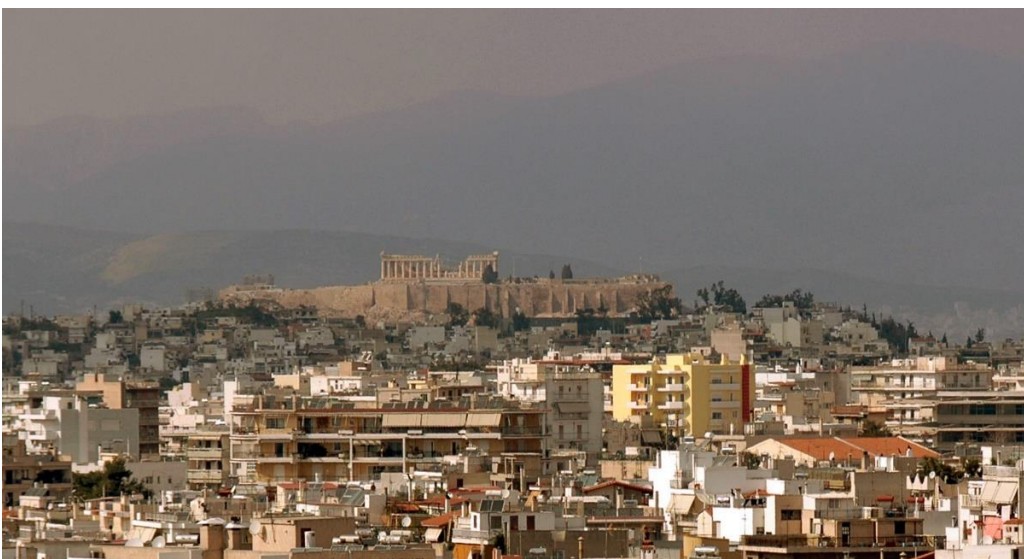

**Figure 1.** The Acropolis hill, an imposing element in the Basin of Athens (Photography: K. Serraos).

## 2. Approach and Methodological Challenges

The archaeological site of the Acropolis of Athens is unique. Thus, it attracts very high local and international visitation. This means risk and emergency management against geophysical and meteorological hazards is important. However, the research and interventions to date on the nexus of monuments and risk concerns are primarily directed toward the safety and protection of tangible monuments [4,16–21].

The uniqueness of the Athens Acropolis dictates severe constraints concerning the character of measures and interventions to be proposed, which need to be balanced with maintaining visitor satisfaction. Interventions should be fully in line with global and national preservation and protection principles and guidelines. Moreover, they should be subject to meticulous approval procedures involving the National Archaeological Council, while structural interventions and construction works should be kept to a minimum. Furthermore, there are noticeable governance challenges in terms of harmonization with civil protection procedures and plans at national, regional, and city levels, while overpassing administrative silos.

The present study is likely the first attempt at more comprehensive emergency planning against geophysical and meteorological hazards in the Acropolis of Athens, one that also considers human safety and protection. Nonetheless, despite severe resource and time limitations, our study needs to be pragmatic enough to provide a founding justification basis for the implementation of the required interventions and guidance regarding emergency planning for other world-class monuments in Greece.

Against this background, our methodology followed sophisticated risk analysis and hazard scenarios that are often considered a sound basis for emergency planning. As a starting point, exposure, vulnerability, and coping capacity are empirically identified and examined with reference to the main physical, human, spatial, and institutional entities/factors that may increase/decrease disaster risks on the site. Then, the main priorities for intervention are identified, including regulating visitor exposure, crowd management, and organizational and governance issues. Hence, pragmatic scenarios for short-term soft interventions are developed to reduce expected impacts on visitors and staff and better manage future emergencies. In addition, a roadmap is produced for long-term risk reduction that will allow for the efficient management of residual risk by means of emergency planning. Clearly, the case at hand could not be tackled with common, widely used risk assessment methodologies and emergency planning approaches. The challenge has been to develop a multidisciplinary approach considering physical, spatial, human, social, and institutional dimensions to realistically reduce human risk and better manage emergencies.

In the following sections, human exposure and vulnerability are briefly discussed in the case study area. Then, emergency evacuation and crowd management are examined through the elaboration of evacuation scenarios based on different hypotheses. Finally, sets of interventions, both structural and non-structural, are proposed, aiming at emergency management and the production of a roadmap for long-term risk reduction (and eventually lower residual risk) manageable by means of emergency planning.

## 3. Visitation as a Proxy for Human Exposure

Human exposure is the most dynamic factor in an intergraded disaster risk assessment and the emergency planning of an archeological site. Here, by human exposure, we mean visitors and staff. In terms of visitors, the overwhelming majority are tourists (from non-Greek-speaking countries), which may include groups of the elderly or students, small guided or independent groups, and individual visitors. Handicapped visitors also have access to the Acropolis hill through the path on the North Slope, which features a specially designed elevator with guard staff.

Human vulnerability is often described through demographic characteristics (e.g., age, gender, ethnicity, education), income level, a lack of social networks and connections, and the risk perception of the exposed population [22–24]. Nevertheless, the vulnerabilities arising through these factors empirically seem to be much less influential in human protection than those pertaining to the space, size, and spatial distribution of the population in the case of the archeological site of the Acropolis. In essence, it is anticipated that high population numbers and density or other high human exposure levels may faster intensify human vulnerability by inhibiting a normal evacuation if large population numbers need to negotiate tight escape routes [25].

The original layout of the Acropolis site, which has only one (narrow) exit/entry available on the hill connected with stairs at Propylaea, creates insecure conditions for human life, even under normal circumstances. The risk of tripping, slipping, or falling on stairs due to their natural wear and tear during use is also substantial. This is in addition to the irregular runways (active/dynamic spaces where the crowd is typically moving), which create uneven flows of visitors and eventually a bottleneck in critical areas, such as the Propylaea, Agrippa monument, and Beule gate (Figure 2). Consequently, visitors may experience crowd-crushing problems (e.g., compressive asphyxia, domino effect) if crowd density or flow exceeds a critical limit in pinch areas during an evacuation procedure. As a matter of fact, high tourism densities produce an interpersonal force that, if left unchecked, can be fatal [26].

A good proxy for human exposure constitutes visitation, namely the number of visitors per average time of visit to the site (2 h). Visitation of the archeological site has steadily been increasing since 2000. In total, 24 million people visited the site in the period from 2011 to 2021, while by far the highest flows were noticed during the tourist season; in fact, 80% of tourists visited the site between April and October over the period in question. Given the visitors' preference for entering the site (west vs. southeast entrance), it was estimated that the average visitation of the hill is four times that of the areas below on an annual basis. This ratio is a comparative measure in relation to the capacity of the space on the hill versus the slopes that helps us focus on the actual visitation on the hill.

The graph cited in Figure 3 illustrates in more detail the daily visitation by time zone on the hill over the period of e-ticketing system operation from June 2018 to September 2021. What becomes evident is that more than 50% of visitors visited the hill during the first two time zones, namely from 8 a.m. to 12 a.m., in the touristic season. This evidence narrows down the time frame of the analysis and indicates the incidence of critical human concentrations in specific time zones (from 8 a.m. to 12 a.m.) and months (from April to October). On average, the daily visitation was around 3000 visitors/2 h on the hill, while on the slopes, it was less than 1500 visitors/2 h in early morning hours, excluding the period from March 2020 onwards, as the visitation sharply declined due to the COVID-19 pandemic and restrictive measures. The frequency of high visitation (>2500 visitors/2 h) on

the hill was considerable during this period; furthermore, 35% of the days of the touristic period experienced high or very high visitation in at least one time zone. However, the most probable scenario may accommodate from 1200 to 2500 visitors/2 h. In effect, more than 80% of the days in the touristic period welcomed this range of visitation in at least one time zone.

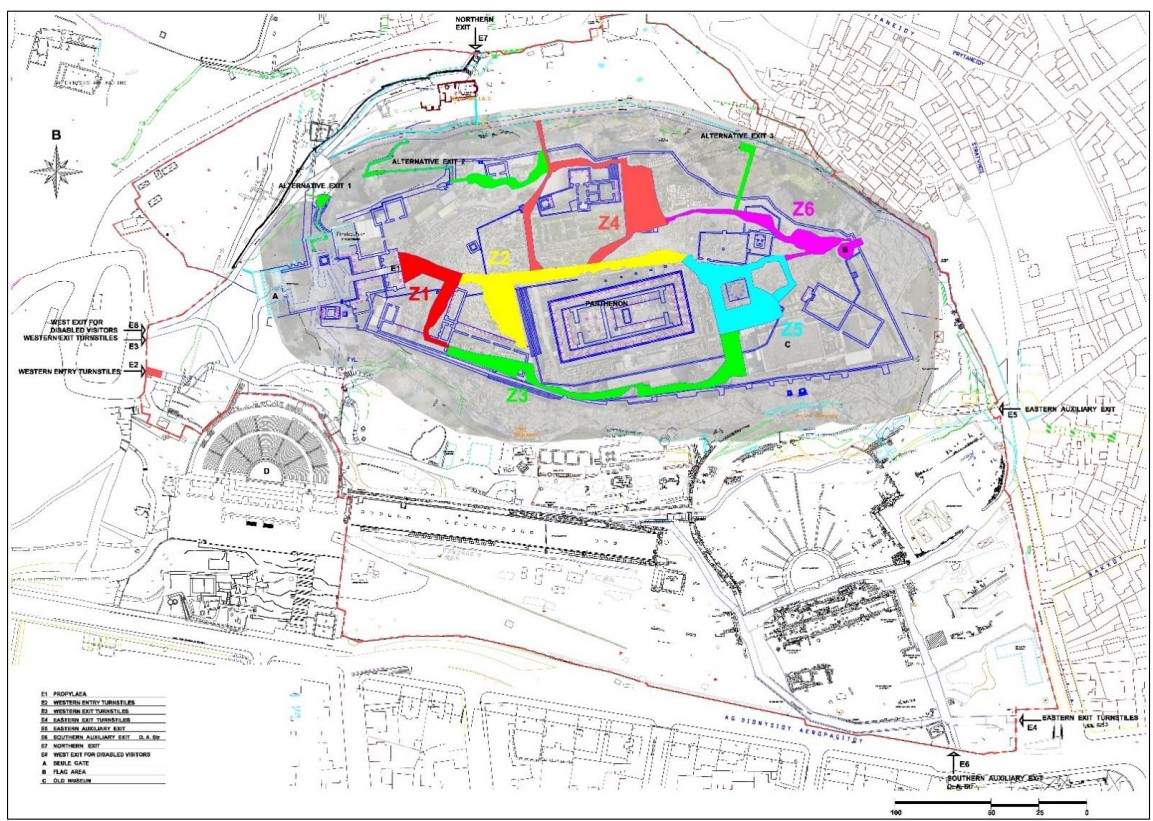

**Figure 2.** Floor plan. Division in evacuation zones. Main access points.

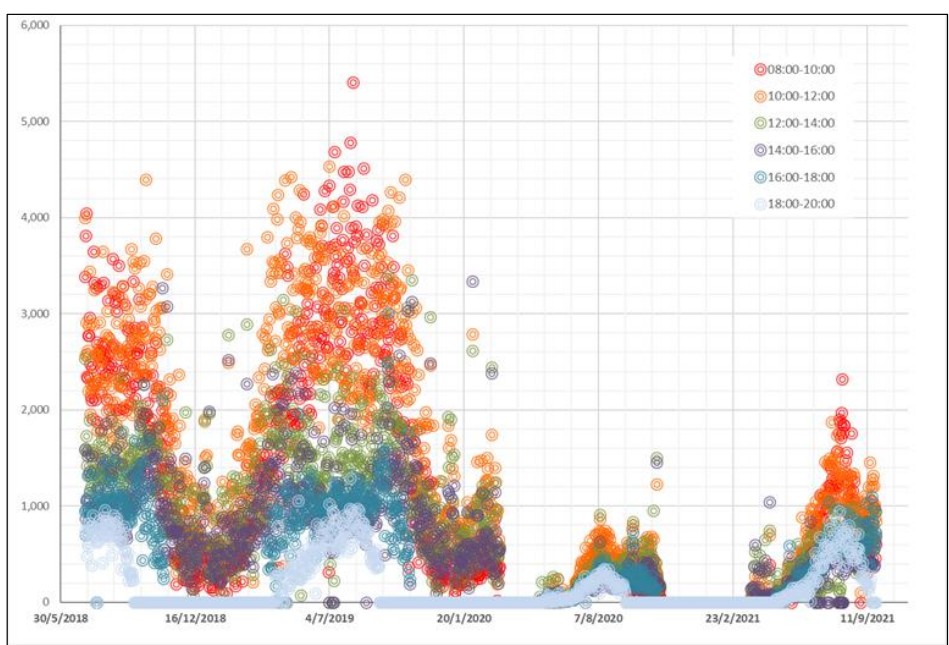

**Figure 3.** Daily visitation by time zone on the Acropolis hill during the period from June 2018 to September 2021 (Source: OADPP, 2021).

## 4. Evaluation of Evacuation Solutions for the Acropolis of Athens in the Event of Hazardous Events

### 4.1. Introduction

Although the Acropolis of Athens is situated in the center of the city, because of its position on top of a rocky hill enclosed by steep walls, it is accessible only from one side (Figure 2). This poses a great challenge to the evacuation of the site.

The purpose of an evacuation plan for the Acropolis of Athens is to manage a safe escape for all people on the site to designated safe areas in the event of a hazardous event (earthquake, fire, landslide, intense rainfall, etc.). The discussion in this section aims to define critical aspects of evacuation planning in the specific site of the Athenian Acropolis.

Taking as the main criterion the prevention of congestion during evacuation [27], we developed, analyzed, and evaluated different scenarios of evacuation processes as the basic research tool of the study. Each scenario represents a different set of spatial and organizational evacuation features in terms of exits to be used, in addition to the type of evacuation (non-guided or guided, concurrently or in a staged process). The evacuation is considered non-guided when the designated employees are not directing the visitors to the appropriate exits.

For each scenario, the main parametrizing factors were the following:

- The number of visitors on the hill at a certain moment in time (population capacity). The estimated number of people in the downhill area was based on the total number of visitors and their average visit time derived from the statistical analysis of ticket sales;
- The walking pace of an average person;
- The behavioral attributes of groups of people.

Each scenario was evaluated based on the results from the simulation. The prevention of bottle-neck phenomena that would force overcrowding in a limited area was the decisive criterion for the evacuation process to be considered acceptable.

### 4.2. Methodology of the Simulation

«Thunderhead Pathfinder software» (TPS) is an agent-based evacuation simulation tool selected to develop the evacuation plan simulation. TPS allows for advanced movement simulation combined with high-quality 3D animated results and has been used to simulate emergency evacuations in case of fires and in closed spaces (Qin et al. 2020, Chen et al. 2020, Shams Abadi 2021). To the best of our knowledge, this is the first time TPS was used to simulate an evacuation in such a unique outdoor environment.

We developed, analyzed, and evaluated different scenarios of evacuation processes. As mentioned above, the main goal and acceptance criterion of the study is the safe escape of all people from the Acropolis with a smooth flow without congestion along the escape routes, especially in the high-risk parts of each route. The study was based on the following assumptions:

- The study examines the entire area of the archaeological site of the Acropolis of Athens, sited within the existing enclosing fence. The existing circulation system remains as it is, consisting of 13,500 m$^2$ of walkways, 4800 m$^2$ of which are on the hill, and 8700 m$^2$ are laid out on the slopes of the hill. The circulation system contains the following points of ingress/egress (Figure 2): Propylaea (E1), the one and only exit of visitors from the fortified hill; two main entries (E2 and E4); two main exits (E3 and E4); two auxiliary exits (E5 and E6); one inactive exit (E7); and one main entry/exit from the hill via elevator for disabled visitors (E8). The varying widths of the walking paths were rounded in their recessed sizes. Additionally, flow-reduction factors such as pavement slipperiness or/and inclinations were included in the selected pace rate and modified TPS accordingly;
- Evacuation begins from the moment a signal/trigger is raised. This signal is assumed to be perceived at the same time by everyone in the study area;

- During an evacuation, no more visitors are allowed to enter the premises. Visitors' entry points are automatically reversed to act as exit points—all doors will automatically be opened in the event of the evacuation signal and will remain open throughout the evacuation process. The passage of visitors through the turnstiles is free and unhindered;
- One-way routes may modify the visitor's flow direction toward the optimal escape route/exit;
- The maximum walking velocity of visitors is set up to 1.0 m/s, and for the disabled and the elderly it is set up to 0.8 m/s. Velocity decreases proportionally on stairs, ramps, and in places with population congestion;
- The use of the elevator is restricted only to people with disabilities. According to the manufacturer's specifications, the elevator's speed is 3.5 m/s. It takes 32 s for a complete vertical movement, and its maximum capacity is two people with disabilities and two attendants.

Based on our knowledge of the actual situation, the examined area was organized into a "downhill area" defined as one unified zone, and an "uphill area" consisting of six discreet populated zones [28] (Z1: Propylaea entry/exit area, Z2: central corridor, Z3: south corridor, Z4: Erehtheion area, Z5: old museum court, Z6: flag area) (Figure 2).

Currently, the "Propylaea exit area" (E1), is the only existing pass for entry/exit to/from the hill of the Acropolis of Athens. Due to its confined space (a dual-lane path with an active width of 1.50 m in each direction, which can offer only a specific flow of people per unit of time, limiting the pace of visitors to a non-satisfactory level), passing through the Propylaea is a slow, risky, and unsafe passage (due to the steep and slippery terrain) in the event of a fast evacuation. Therefore, the number of visitors on the hill proved to be the most crucial factor for the evacuation process.

Considering the collected data on the minimum/average/maximum number of visitors during a sequence of yearly seasons (Figure 3), a baseline scenario was needed to set the capacity of the site to securely manage emergency situations and evacuation capabilities. The Nr of 500 visitors was selected as the baseline, whilst evacuation simulations were executed for up to 2500 visitors.

*4.3. Simulations*

4.3.1. Scenario 1

Scenario 1 is the baseline and reflects the current situation if no soft or hard interventions are performed. In addition to the general assumptions presented in part 4.2, the main assumptions in Scenario 1 were: [29]

- Visitors move toward the exits on their own initiative and without guidance by designated employees;
- Each visitor moves freely in space and looks for exits on his/her own;
- Visitors on the hill of the Acropolis of Athens are evacuated through the Propylaea (E1) exit, which is the only existing evacuation route.

In Scenario 1, the total number of visitors is 730, of which 530 are in the uphill area. Among visitors in the uphill area, 30 are visitors with disabilities and their attendants.

Based on the simulation results:

The maximum walking distance was performed by a visitor [V1] initially located in the "flag area" (Z6—point B—Figure 2), and they headed westward through the Propylaea (E1) to the western entry (E2). V1 performance was as follows: estimated walking distance = 447 m, time spent = 461 s (7.7 min), and average walking speed = 0.97 m/s. However, the maximum evacuation time was spent by a visitor [V2] initially located in the "old museum court area" (Z5—point C—Figure 2), who also headed westward through the Propylaea (E1) to the western entry (E2). V2 performance was as follows: estimated walking distance = 425 m, time spent = 590 s (9.8 min), and average walking speed = 0.72 m/s.

Visitor V1 almost reached maximum speed (1.0 m/s). He/she did not encounter any obstacles to delay his/her exit. On the contrary, visitor V2 only reached an average speed of 0.72 m/s and encountered congestion near the Propylaea (E1). This differentiation is due to the different times they arrived at the Propylaea exit area. V2 was closer to the Propylaea (E1) and arrived earlier than V1, who started his route from the easternmost point of the Hill, the "flag area" (Z6—point B). When V1 arrived at the Propylaea (E1), the congestion was resolved, and the visitors' motion had normalized. The motion of the disabled visitors did not show any congestion.

An unexpected "exit preference pattern" was observed whereby 528 out of 700 + 30 visitors "preferred" to leave the area through the E2—Western Entry, which is the same point from where they entered the archeological site. As is documented in the simulation video, they adhered to the "Logic of the Crowd" [29]; that is, they followed the person ahead of them (Figure 4). Since there was no congestion, they did not look for alternative routes (Table 1).

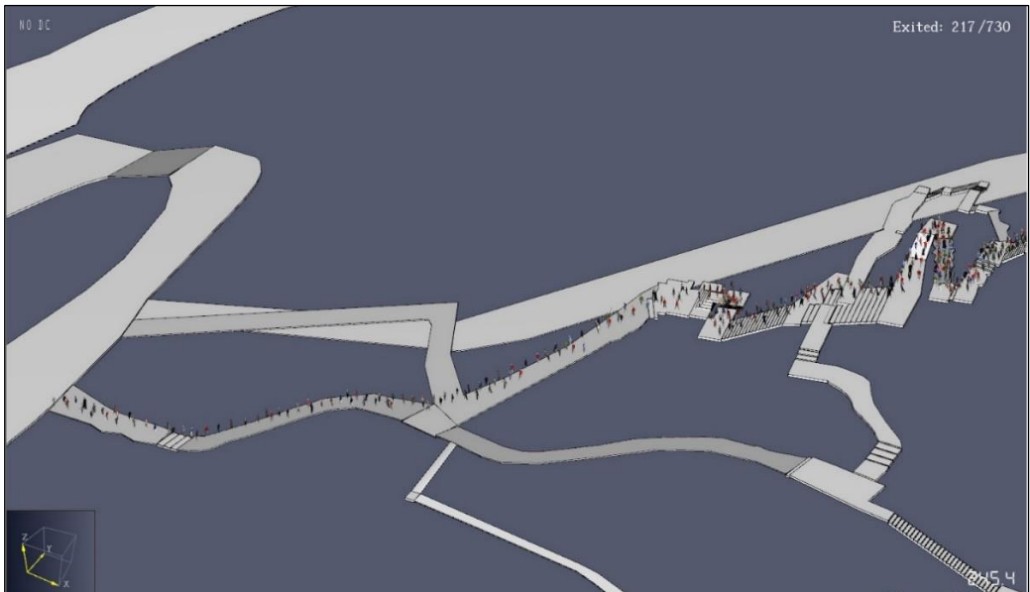

**Figure 4.** Crowd logic—exit from the western entry turnstiles.

**Table 1.** Estimated exit usage and passing rate in visitors that must be safely evacuated without any guidance.

| Exit | Usage Rate (nr of Passing Visitors) | Passing Rate (nr of Visitors Per s) |
|---|---|---|
| E1—The Propylaea | | |
| E2—Western Entry/entry turnstiles | 528 | 0.31 |
| E3—Western Exit/exit turnstiles | 6 | 0.13 |
| E4—Eastern Exit/exit turnstiles | 5 | 0.23 |
| E5—Eastern Auxiliary Exit | 31 | 0.12 |
| E6—Southern Auxiliary Exit/D. A. Str. | 65 | 0.30 |
| E7—Northern Exit/presently inactive | 45 | 0.18 |
| E8—Western Exit for the Disabled | 50 | 0.06 |
| Totals | 730 | |

Conclusion: Scenario 1 was an attempt to determine the maximum possible number of visitors on the hill of the Acropolis of Athens that can be safely evacuated through the only existing evacuation route and exit Propylaea (E1). The archaeological area around the hill does not present any significant problems due to many alternative evacuation routes and exits. The experiment proved that a crowd of approximately 500 visitors can be safely

evacuated from the hill without any guidance, on time, and at a regular pace rate. However, Scenario 1 provides for only a small portion (500 people) of the real number of visitors on the site and is only considered as a base of comparison for the following experiments.

### 4.3.2. Scenario 2

In Scenario 2, evacuation is guided by designated employees. Evacuation from the uphill area is scheduled into six consecutive steps based on the six-zone spatial organization (Figure 2). Evacuation of each zone requires the complete evacuation of the adjacent one. Visitors should be kept within the boundaries of each zone. In the simulation, Z1 was the first to be evacuated, being the closest to the Propylaea (E1). Z2 was the second to be evacuated as it is the nearest to Z1. Z4, Z3, Z5, and Z6 followed.

An alternative exit route (aE1) is added. The (aE1) exit route is located to the north of the monument of Agrippa (Figures 2 and 5). This new route helps decrease the congestion of people observed in the evacuation route between the Propylaea (E1) and the "Beule Gate" (Point A—Figure 2), an area of particular risk due to its long and steep inclination and slippery terrain. In Scenario 2, the total number of visitors is 1530, of which 1230 are in the uphill area. Among visitors in the uphill area, 30 are visitors with disabilities and their attendants.

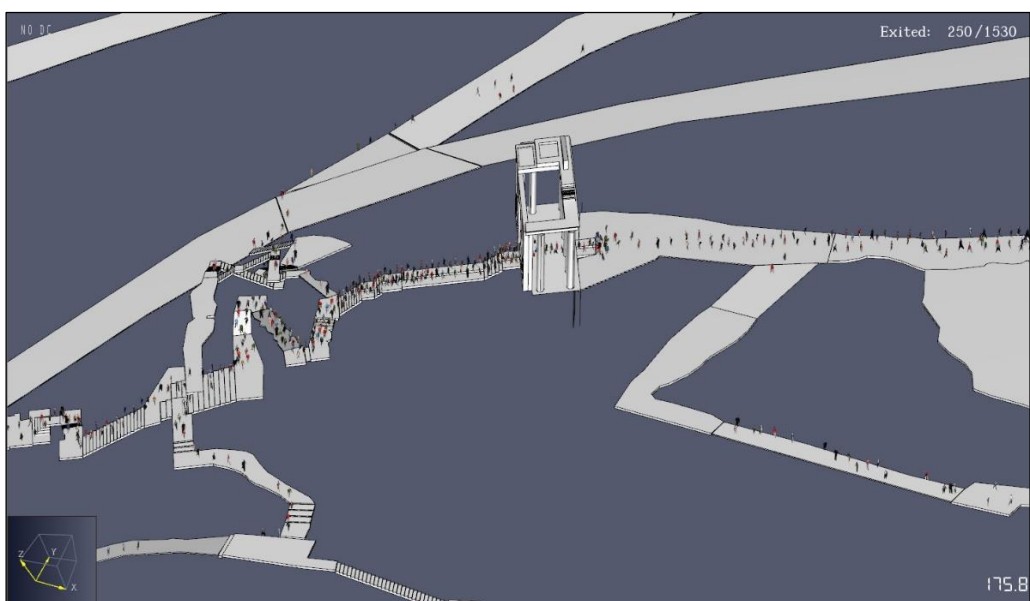

**Figure 5.** Access from the Propylaea. Zone 2 is currently being evacuated.

Conclusion: Scenario 2 allows for a safe evacuation of a larger number of visitors; however, the number of visitors remains relatively small compared with real visitation data. For this solution to be implemented, an alternative exit route (aE1—Figure 2) must be opened and used. Moreover, an emergency plan must be in place and implemented successfully.

### 4.3.3. Scenario 3

Scenario 3 modifies Scenario 2 by increasing the number of exits. In addition to the use of the alternative exit route (aE1), another alternative exit route (aE2) is added to the northwest of Erechtheio (Figures 2 and 6) where there is the possibility to erect an alternative exit staircase, of temporary construction, following the steeples formation of the hill at the point. This alternative route (aE2) aims to relieve the evacuation procedure of the uphill area or/and provide an emergency exit in case of a loss of access to the Propylaea exit (E1), i.e., due to a structural collapse. The proposed (aE2) exit route was used in antiquity but today remains inactive. Additionally, in this Scenario, evacuation of each zone requires the complete evacuation of the preceding area. In the simulation, the order of zones to be evacuated is Z1, Z2 Z4, Z3, Z6, and Z5.

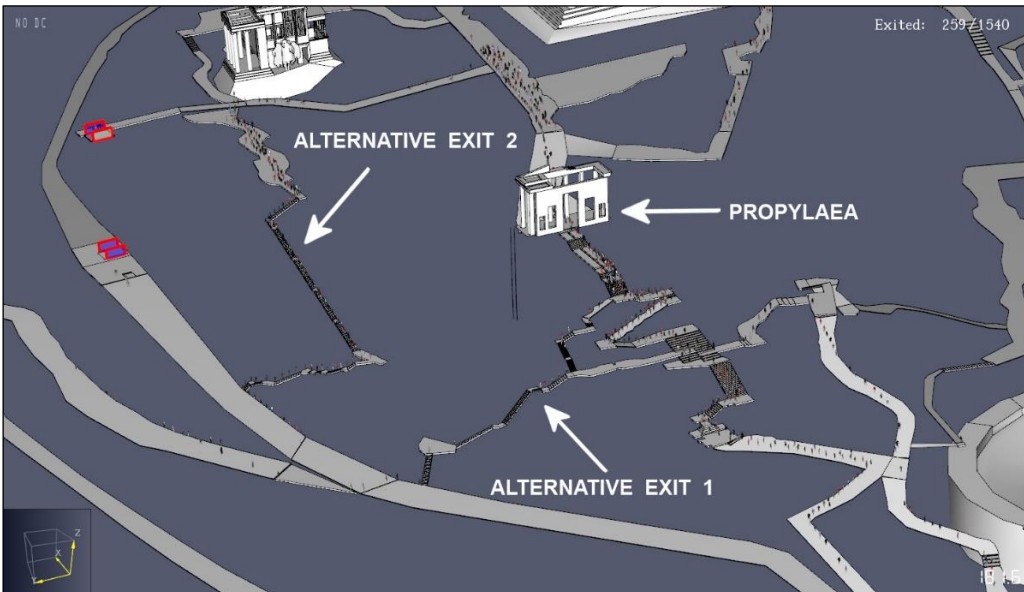

**Figure 6.** Alternative exit 2 appears on the left and Alternative exit 1 on the right.

The number of visitors considered in this Scenario is 1530, of which 1230 are on the uphill area. Among visitors in the uphill area, 30 are visitors with disabilities and their attendants.

Conclusion: Scenario 3 was an attempt to further improve the evacuation procedure by providing an additional evacuation route (aE2) from the uphill area. In this case, the maximum evacuation time was reduced by 88 s (1.5 min) due to the reduction in waiting time for visitors before beginning evacuation. The alternative second exit route from the uphill area contributes to safety and provides substantial relief to the risky and stressful exit route through the "Prolylaea–Beule Gate" (E1) part. Overall, Scenario 3 is satisfactory; however, the number of visitors remains relatively small compared with the real visitation data.

### 4.3.4. Scenario 4

Scenario 4 is a modification of Scenarios 1 and 3. As in Scenario 1, only the uphill exit through Propylaea is used. However, a six-step guided evacuation procedure based on the six-zone spatial organization of the uphill area is performed.

As in Scenario 3, visitors should be kept within the boundaries of each zone. Evacuation of each zone requires the complete evacuation of the preceding one. In the crowd motion simulation model, Z1 was the first to be evacuated, being the closest to the Propylaea (E1). Z2 was the second to be evacuated as it is the nearest to Z1. Z4, Z3, Z6, and Z5 followed.

The number of visitors considered in this scenario is 3330, of which 2530 are in the uphill area. Among visitors in the uphill area, 30 are visitors with disabilities and their attendants.

Based on the simulation results:

The maximum walking distance was performed by a visitor [V1] initially located in the "flag area" (Z6) who headed westward through the alternative exit (aE1) in the north (E7). V1 performance was as follows: estimated walking distance = 537 m, time spent = 873 s (14.55 min), and average walking speed = 0.61 m/s. Due to V1's placement in the furthermost Z6 area, the overall evacuation time consumed (including a waiting time of 570 s needed before all other zones were consecutively emptied) was 1443 s (24.05 min). The maximum evacuation time was consumed by a visitor [V2] initially located in the "flag area" (Z6–Point B—Figure 2) also headed westward through the Propylaea (E1), then to the "Herodian theater route" (Point D—Figure 2), and to the western entry (E2).

V2's performance was as follows: estimated walking distance = 487 m, time spent = 933 s (15.55 min), and average walking speed = 0.52 m/s. In total, including an initial delay of 570 s, V2's evacuation consumed 1505 s (25.1 min), which is 10.1 min more compared with Scenario 3's same route.

Both V1 and V2 presented a low walking speed. As the simulation videos show, in the Propylaea exit area (E1), they both encountered significant congestion and remained motionless for several seconds until they were able to pass through toward the following exits. Congestion density in the Propylaea exit area (E1) was quite intense, providing almost six motionless people per m² (Figure 7), which renders this scenario practically unacceptable. The congestion phenomenon in the Propylaea area (E1) affected the motion of the disabled and elderly people, creating obstacles to the use of the elevator. On average, they traveled 430 m in 676 s, and their average walking speed was 0.63 m/s.

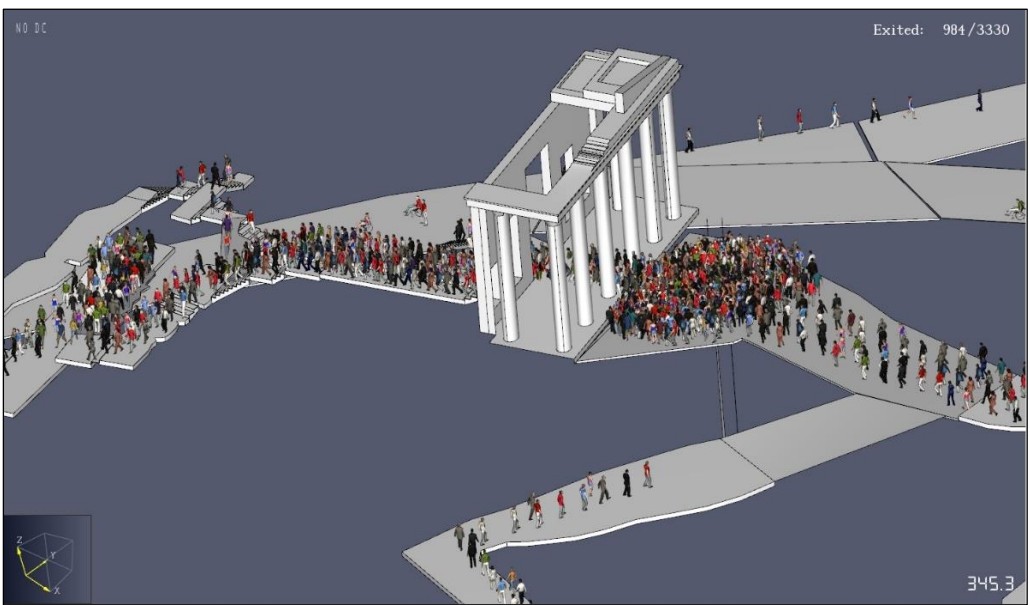

**Figure 7.** Increased congestion in the Propylaea.

Conclusion: Scenario 4 was an attempt to safely evacuate a larger number of visitors compared with Scenario 3, under the provision that all visitors exit the uphill areas through the Propylaea exit (E1). Employing a six-staged evacuation method from the hill was not enough to offer a safe evacuation to such a large population (2500 people). The increased demand for access through the Propylaea–Beule Gate route in combination with the long delays in starting the evacuation in each zone projected a scenario that poses serious risks to the integrity of the visitors [29].

4.3.5. Scenario 5

Scenario 5 is an attempt to improve Scenario 4, which was not successfully resolved. However, in the present scenario, both alternative exit routes (E1/Beule Gate or aE1) and (aE2) (Figure 2) are in use. Again, evacuation is guided and takes place in six steps. Visitors should be kept within the boundaries of each zone. Evacuation of each zone requires the complete evacuation of the preceding one. In the crowd motion simulation model (TPS), Z1 and Z4 were the first to be evacuated, being the closest to the Propylaea exit (E1) and to the alternative NW exit route (aE2), respectively. Z2 was the second to be evacuated as it is the nearest to Z1 and to Z4. Z3, Z5, and Z6 followed (Figure 8).

The number of visitors considered in this scenario is the same as in Scenario 4; that is, 3330, of which 2530 are in the uphill area. Among visitors in the uphill area, 30 are visitors with disabilities and their attendants.

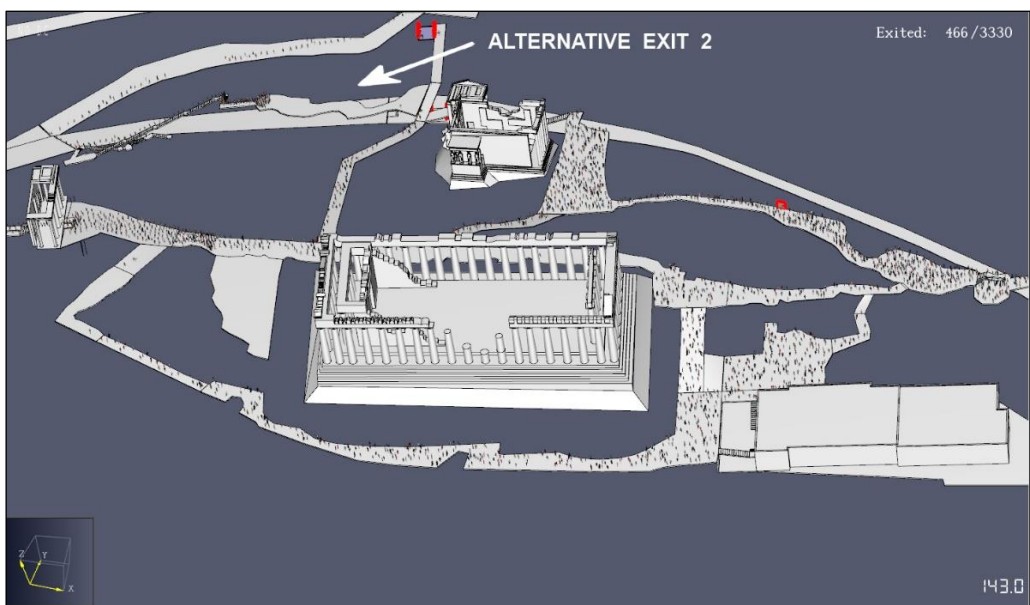

**Figure 8.** Evacuation in stages. Zone 1 has been evacuated; Zone 2 is currently being evacuated.

Conclusion: The results were marginally acceptable, with the most unfavorable point being the small congestion in the Propylaea exit (E1), which, however, lasted for a short time. The main prerequisites for the successful implementation and operation of Scenario 5 were the opening and use of alternative exit routes (aE1 and aE2—Figure 2) and the establishment and implementation of an emergency evacuation plan.

### 4.3.6. Scenario 6

Scenario 6 is based on the same characteristics as Scenario 5. However, besides both alternative exit routes (E1/Beule Gate or aE1 and aE2) (Figure 2), a third one (aE3) is added to the northeast of Erechtheion in the area where an existing natural gulch in the rocky formation of the hill crosses the fortification Walls (Figure 2).

Like in previous scenarios, visitors should be kept within the boundaries of each zone. Evacuation of each zone requires the complete evacuation of the preceding one. In the crowd motion simulation model (TPS), Z1, Z4, and Z6 were the first to be evacuated, being the closest to the Propylaea exit (E1), the alternative NW exit route (aE2), and to the new alternative exit (aE3), respectively. Z2 was the second to be evacuated as it is the nearest to Z1 and to Z4. The Z3 and Z5 areas were evacuated from all exits.

The number of visitors considered in this scenario is the same as in Scenario 4; that is, 3330, of which 2530 were in the uphill area. Among visitors in the uphill area, 30 are visitors with disabilities and their attendants.

Based on the simulation results:

The maximum walking distance was performed by a visitor [V1] initially located in the "old museum court" (Z5–Point C—Figure 2) headed northward through the new alternative exit (aE3) to the Eastern Auxiliary Exit (E5). V1's performance was as follows: estimated walking distance = 568 m, time spent = 751 s (12.5 min), and average walking speed = 0.76 m/s. Due to V1's placement in the furthermost Z5 area, the overall evacuation time spent (including a waiting time of 500 s needed before all other zones are consecutively emptied) was 1251 s (20.85 min). The maximum evacuation time was spent by a visitor [V2] initially located in the "old museum court" (Z5–Point C—Figure 2), who also headed northward through the new alternative exit (aE3) to the Eastern Auxiliary Exit (E5). V2's performance was as follows: estimated walking distance = 506 m, time spent = 760 s (12.7 min), and average walking speed = 0.66 m/s. In total, including an initial delay of 500 s, V2's evacuation took 1260 s (21 min), which is 1.7 min less than Scenario 5's longer route. Both V1 (0.76 m/s) and V2 (0.66 m/s) presented a lower walking speed than expected,

which can be attributed to the high elevation difference at the point where the new NE alternative exit (aE3) was placed, and not due to congestion. The motion of the disabled and elderly visitors who used the elevator did not show any congestion. The average rate of exiting visitors crossing Propylaea exit (E1) (Figure 9) was approx. 0,90 persons/s compared to 0.97 p/s in Scenario 5, which proves, again, the positive performance of the evacuation through the NE alternative exit (aE3).

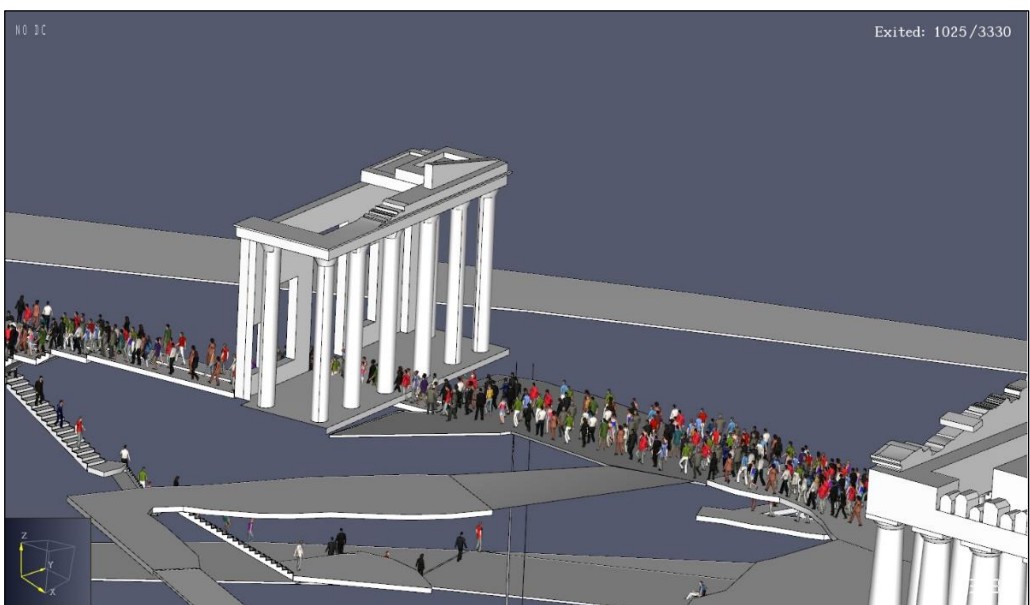

**Figure 9.** Access rate through Propylaea.

"Exit preference pattern" has been also observed, as 1127 out of 3330 visitors "preferred" to leave the area through the Eastern Auxiliary Exit (E5).

Conclusion: In this case, the results were satisfactory. A reduction in both the maximum and average evacuation time was observed. This scenario is also equipped to handle a potential collapse of the Propylaea passage because a large and stable emergency exit that allows for the evacuation of the top of the hill was added. The main prerequisites were the opening and use of the alternative exit routes (aE1 and aE2—Figures 2 and 10), the creation of a new alternative exit, (aE3—The NE staircase alternative internal route), as well as the establishment and implementation of an emergency evacuation plan. It should be noted that the location and specifications of the NE alternative exit (aE3), as well as its potential intervention in the Acropolis monument, should be the subject of a future study.

*4.4. Conclusions from the Simulations*

Using Thunderhead Pathfinder software, we developed, analyzed, and evaluated different scenarios of evacuation processes, each representing a different set of spatial and organizational evacuation features (Table 2).

From the aggregated results of all examined scenarios, we identified critical aspects for the evacuation planning of the Acropolis of Athens.

Based on visitors' population data analysis and on simulation model results, "downhill" areas do not show any significant problems (no congestion observed) during the evacuation, mainly due to the several available exits (7) on the existing fencing. As simulation models showed, congestion phenomena appeared on several occasions of evacuation from the "uphill area". Most congestion occurred in the "Propylaea exit area" (E1), since it is (as of today) the only existing pass for entry/exit to/from the hill of the Acropolis of Athens. Without other interventions or guidance to manage the evacuation, only small numbers of visitors (up to 500 or slightly more) on uphill areas can be safely evacuated (Scenario 1).

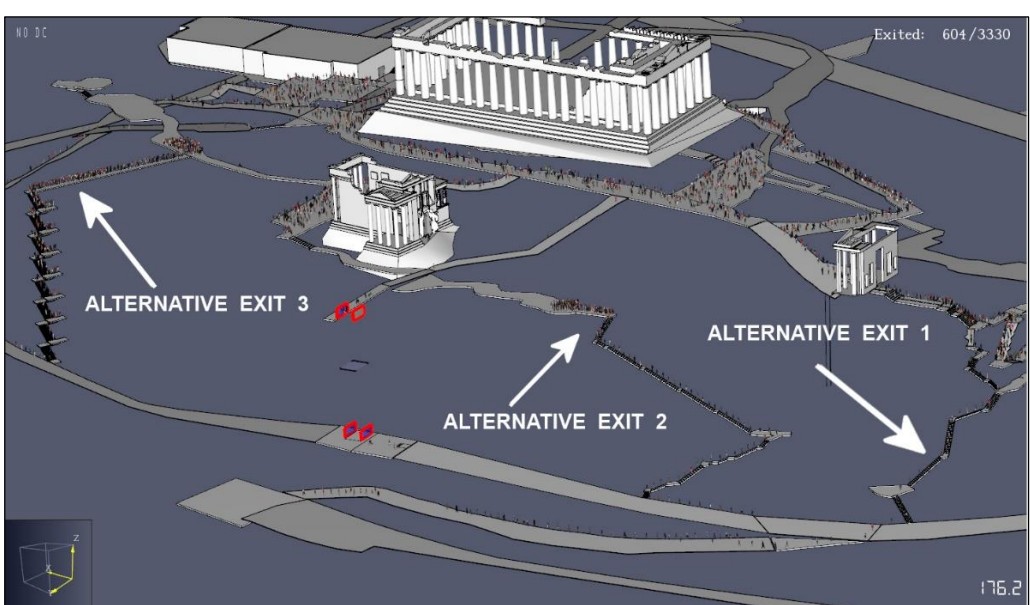

**Figure 10.** Alternative exits 1,2, and 3.

**Table 2.** Simulated scenarios and their aggregated results.

| Scenarios | 1 | 2 | 3 | 4 | 5 | 6 |
|---|---|---|---|---|---|---|
| *Evacuation exits* | | | | | | |
| Six-step guided evacuation | - | Yes | Yes | Yes | Yes | Yes |
| Uphill exit route: E1/Beule Gate | Yes | Yes | Yes | Y | Y | Yes |
| Uphill exit route: aE1 (Agripa M.) | - | - | Yes | - | Yes | Yes |
| Uphill exit route: aE2/NW staircase | - | - | Yes | - | Yes | Yes |
| Uphill exit route: aE3/NE staircase | - | - | - | - | - | Yes |
| *Number and distribution of visitors* | | | | | | |
| Nr of Visitors—uphill | 500 | 1200 | 1200 | 2500 | 2500 | 2500 |
| Nr of Visitors—downhill | 200 | 300 | 300 | 800 | 800 | 800 |
| Nr of disabled and elderly people | 30 | 30 | 30 | 30 | 30 | 30 |
| *Results from the simulation* | | | | | | |
| Maximum walking distance (m) | 447 | 502 | 489 | 537 | 540 | 568 |
| Maximum evacuation time (s) | 590 | 576 | 486 | 690 | 614 | 554 |
| Maximum total evacuation time (s) | 590 | 986 | 898 | 1443 | 1330 | 1251 |
| Average walking speed (m/s) | 0.81 | 0.61 | 0.64 | 0.44 | 0.50 | 0.51 |
| Evaluation of evacuation procedure (worst = 0/1/2 = best) | 2 | 1 | 1/2 | 0 | 1 | 2 |

If evacuation is guided and managed, a six-step-like procedure of consecutively evacuating six zones can result in doubling the numbers of visitors that can be safely evacuated from the hill to more than 1000–1200, as in Scenario 2. However, congestion phenomena may not be avoided, and for that reason, an additional alternative exit route (aE1) is suggested. As shown in simulation models, this may improve the evacuation procedure significantly, as in Scenario 3.

Scenarios 4, 5, and 6 attempt to examine the evacuation procedure for the numbers of visitors that approximately visit the monument, and they simulate real conditions in the uphill areas based on visitor population data statistics. As shown in simulation models, by doubling the number of visitors on the hill up to 2500, the existing spatial and traffic organization cannot bear the safe evacuation of all people. Congestions are at their peak and evacuation time becomes endless, whether a guided procedure is available or not, as in Scenario 4. The provision of alternative exit routes (aE1, or/and aE2, or/and aE3)

to accommodate the evacuation procedure and relieve the Propylaea exit area (E1) from intense congestion is needed. Scenario 5 shows that the use of the NW alternative exit (aE2) can make a difference in safety during evacuation; however, congestion phenomena kept occurring. Scenario 6 strives to form a more ambitious but reliable management plan of safe evacuation for all people under real conditions by proposing the addition of a third alternative exit route to the NE (aE3). Although walking speeds are considerably reduced compared with those projected in Scenarios 1–3, no congestion was found in the simulation models, which confirms that realistic planning for a safe evacuation is achievable.

Because of the above comments, we can conclude that for a number of 2500 or more visitors on the uphill areas, no safe evacuation can take place unless a third alternative exit route from the hill is established (aE3). The characteristics and location of this exit should be the result of a separate study. The maximum number of visitors on the hill will, in this case, be proportional to the capacity of this new main exit.

Furthermore, the evacuation conditions in the uphill areas must be improved. The irregular width of the walking paths, slippery terrain, steep inclination in some areas, narrow passages in many places, increased visitor traffic that creates "bottle-neck" phenomena, and the lack of sufficient and durable protective elements from falling make the visit along the hill a potentially risky experience, especially when these factors occur together with the development of a hazardous natural phenomenon.

## 5. Directives and Instructions on Proposed Infrastructure Projects, Planning, and Implementation

This section attempts to formulate a framework of proposed infrastructure projects aiming to tackle the risks faced by people to manage the process of evacuation from the archaeological site of the Acropolis of Athens. The inspected site, apprenticed as a world-class monument and a prime cultural asset of unique "value", dictates a holistic planning approach so that each goal may achieve the best results in all critical aspects, i.e., sustainability, high environmental efficiency, and provision for the safety and health of people, regardless of any emerging dangerous phenomena.

The planning strategy of any proposed infrastructure project should be based on an inventory of design criteria and principles deriving from the particularities of the site and the qualitative characteristics of the project scope, such as the uniqueness of the geomorphology and the spatiality of the archaeological landscape of the Acropolis of Athens and its central position within the urban fabric of the city of Athens. Additionally required is the aesthetic integration and harmonization of all proposed infrastructures to the architectural and historical context of the Acropolis, and the reversibility of constructions by delivering innovative solutions and mobilizing advanced technology to prevent any changes to the monument. In a more general perspective, all interventions should provide solutions toward a sustainable and resilient life cycle and ensure the health and safety of people during construction and operation; furthermore, any implementations should be interoperable and manageable by the available staff [30–32].

Regarding infrastructures in such a demanding site, it is of great importance to ensure the efficiency and effectiveness of interventions, functional reliability, adequate operational lifespan, and the possibility of expanding or upgrading the installed systems without requiring new works or without causing any damages to infrastructure or monuments.

Following the above criteria, infrastructure projects can be grouped into the following categories depending on their time of activation: infrastructures that operate preventively and managerially; those that aim at the proper operation and supervision of the site, evaluate the situation at any given time, and estimate risk; those that provide warnings before the occurrence of dangerous phenomena; those that operate during the occurrence of a dangerous phenomenon, or any other related case of risk, and are called to manage the process of people's evacuation in a timely and safe manner; and those that operate in the context of the restoration of the site immediately after the occurrence of a dangerous phenomenon [19,20] by assessing the operating status of the systems and applying pre-

designed protocols for the management of damage, breakdowns, injuries, etc. Continuous and reliable coordination among all of the above is critical.

Finally, while all infrastructure projects should be responsive to any type of risk scenario examined in the present study, they should also provide flexibility and adaptability in the case of a redesign or the addition of new ones. All interventions should comply with the entirety of the regulatory frameworks that govern these infrastructure projects, which consist of regulations, technical instructions, specifications, and standards, which are valid and applicable in Greece and/or the European Union.

As a result of the analysis in previous sections, the creation of comprehensive planning for infrastructure projects to manage the risks in the Acropolis of Athens, and the implementation of targeted interventions that will serve the purposes of this planning, emerged as an urgent necessity.

The basic conditions and requirements that such planning must meet, indicatively but not restrictively, should include the following: the application of an accurate and reliable registry, a recording and coordination system for all existing or planned infrastructures (premises, networks, and equipment/facilities, etc.), and the utilization of a BIM digital environment for the entire area, which will include all of the area's required information and documentation data. The BIM platform should act as a reliable interface for exchanges of information; however, it must be compatible with the existing systems of control and provide a real-time recording of the site, as should any other system of monitoring, management, and maintenance for the premises installed in the future.

The design and application of the "emergency-responsive and short recovery plan" (ERSRP) for the Acropolis of Athens, should include in its core an evacuation management plan specializing in aspects of the evacuation process, i.e., the traffic organization of people and things, the distribution of safe zones, area capacities, primary and secondary escape routes, exits, and the qualitative specifications of all constructions, such as driveways, automatic exit doors, and signage. For the support of the planning, critical upgrades in existing infrastructure and/or the addition of new constructions are necessary. Indicatively but not restrictively, the most urgent ones are: (a) the upgrading of fencing, demarcation, and signage systems of all archaeological areas, restoring the operation of all ingress/egress, the rehabilitation of traffic-circulation routes providing accessibility for all people, and (b) the reorganization of safe-zone areas in terms of their geometric and qualitative characteristics. Restoration of interoperability and automated control over critical risk-management systems, such as evacuation exits, etc., is also important, as is the option of providing an additional evacuation staircase on the Acropolis of Athens. Lastly (c), there should be the creation of an area and facilities management and control center, a temporary shelter for disabled individuals during severe weather phenomena, an emergency first-aid health center, and the upgrading of elements and means for the protection of all people from falling, collision, and impact. The creation and operation of a facilities maintenance center for stocking provisions of spare parts and critical components of infrastructure for their immediate maintenance and/or repair should also be a part of the above infrastructures.

In addition to the above, the provision of a cloud-based project-closure file in coordination with the safety and health plan compatible with a future facilities management and maintenance operation system for all infrastructure projects on the site is absolutely necessary. The BIM platform is a possible host of such an archive and shall include at least the following assets: as-built drawings and implementation data of all infrastructure, technical instructions and specifications of construction, operation and maintenance manuals of all systems, key suppliers, and maintainers, and a facilities maintenance book with a registry for the recording of defects and damages to the systems.

Critical existing technical systems (TS) provided on the Acropolis that need an upgrade and/or the provision of new ones are categorized: (i) infrastructures and installations that regulate the overall operation of the site and (ii) infrastructures and installations that are specialized to confront specific phenomena, e.g., lightning protection or seismic events. Indicatively, such (TS) systems are (1) water management infrastructure (potable

water supply, irrigation, recycling, storm-water drainage, etc.), (2) waste and recycling management, with an automated system for recording and managing waste types on-site, (3) means for the production, supply, and management of energy, (4) wireless communication, security, and alarm systems, (5) safety and evacuation signage and information systems, (6) aerial surveillance systems for transmitting video and audio information from the site, (7) fire detection and extinguishing systems, (8) crowd management systems that record and capture information regarding the population and its distribution in the site in real-time, and (9) automated monitoring and management systems for all facilities, located in the control center, that is compatible and interconnected with all the aforementioned subsystems. The interoperability and interconnectivity of the technical systems are a condition for their reliable operation, management, and maintenance, and therefore for the successful implementation of the ERSRPlan.

## 6. Discussion: Proposals Concerning Critical Issues That Call for Solutions

The present study is the first of its kind for the Acropolis of Athens; therefore, it mainly identifies issues that need to be dealt with after further consideration at the next stage. A main conclusion of the study is that, mainly due to unregulated high visitation, the risk of geophysical and meteorological hazards is too high to be tackled solely by means of emergency management. What is needed is a comprehensive strategy for risk reduction so that residual risk becomes eventually manageable through effective crowd management and emergency planning. Furthermore, even when the safety of visitors is set as the supreme priority, how visitation can be regulated without compromising the rights of visitors to access a world heritage monument remains a challenging governance issue.

To this end, the outcome of the study is not an evacuation plan per se but sets of proposed short-, medium-, and long-term interventions, both soft and hard, for various areas, from disaster risk reduction to emergency management and capacity building to policy making and governance. In the following, we briefly present sets of proposed interventions aimed as a basis for decision making and governance.

### 6.1. Preventive Measures and Actions to Reduce Risk

From the various evacuation scenarios tried, which were based on the criterion that there is no congestion during evacuation, it was found that at present, the maximum number of visitors on Acropolis Hill should not exceed 500 people. This can be increased to a maximum of 2500 people provided that structural and organizational interventions are introduced. Consequently, given the reported high visitation rates, arrangements must be introduced to regulate visitation. As a starting point, constant monitoring of the number of visitors by means of electronic control of incoming and outcoming visitors will allow for keeping the number of visitors below maximum. For this, an expansion and upgrade of the tourniquet system and the introduction of additional devices for measuring the flow of visitors are required.

Additionally, to smooth visitation peaks, a new pricing policy based on different ticket prices per time slot must be introduced. The new policy will be more effective if booking management (online, through travel agencies, and in person) is improved to support keeping the number of tickets per time slot up to a maximum. To this end, communication and collaboration with various stakeholders are essential.

Vulnerability of visitors is also a matter to be considered. Visitors with characteristics of increased vulnerability should be warned to avoid visiting the monument when conditions are expected to be harsh for them. For example, when weather is very hot for elderly people or when slipperiness is increased due to rain or frost for people with mobility difficulties. In any case, useful information must be provided to all visitors about self-protection measures (e.g., recommendations for appropriate clothing, footwear, and equipment) relating to the prevailing conditions at the time of their visit. In addition, employees should follow respective appropriate self-protection measures.

### 6.2. Spatial and Operational Issues

The analysis of the special characteristics of the area revealed malfunctions and dangers, leading to the need for interventions in several critical fields.

For one, the network of visitation routes should be better designed so that it facilitates visitors' spatial awareness and prohibits visitors from accessing high-risk areas (e.g., due to landslides) [33]. A matter to be considered is that currently, routes serving the ongoing construction works on the hill are the same as the ones the visitors use. Moreover, existing physical barriers, both natural and artificial, should be removed, and suitable, stable, non-slip flooring should be placed on the main routes. As a short-term intervention, signage in the archeological site needs to be better designed and upgraded, giving more emphasis to the safety of visitors.

At present, the main entrance–exit of visitors to Acropolis Hill is Propylaea, and therefore special consideration should be given to how to assess and appropriately reduce risk at this specific point. This task is far from easy but it is a significant one, as a failure there may block exit from the hill. To this end, the planning of an alternative and/or additional entrance–exit for Acropolis Hill, placed in a suitable location so as to be as unintrusive as possible, should be considered. This again is a major governance challenge because stakeholders with diverse perceptions, interests, and priorities are involved, with individuals from various disciplines, archeology, tourism, health and safety, and civil protection, having a say on the issue [34].

### 6.3. Planning and Preparation for Dealing with Emergencies

It is of utmost importance to elaborate on an emergency plan, including an emergency evacuation plan, of the Acropolis of Athens as soon as possible, and to proceed with all measures required to make this plan operational and actionable. Among these measures, an advanced yet critical one should be the establishment of an emergency and control center that could support real-time monitoring of the whole of the archeological site in normal conditions as well as early warning and emergency response. Furthermore, a system of emergency notification and warning should be implemented so that visitors and staff at the archeological site are informed and guided in case of an emergency.

In addition, the role of the guard team is considered crucial in the case of an emergency. The guards should be dressed in uniforms to be easily identified and distinguished from the crowd while carrying all the necessary technical equipment for emergency management.

### 6.4. Capacity Building—Education and Training

The effective response of the staff in a crisis requires constant training through lifelong learning structures. Such training can be provided by the National Center for Public Administration and Local Government and, more recently, the higher education institutions of the country.

In addition, other stakeholders involved in the operation of the archaeological site (e.g., tourist guides and leaders of tourist groups) are also eligible for systematic training on emergency management, and it is proposed that relevant informative and educational activities should be organized on the initiative of the Ministry of Culture and Sports.

### 6.5. Governance Issues

According to current civil protection legislation, the Ministry of Culture and Sports is not among the agencies that are legally obliged to have an emergency plan in place. However, the Ministry is responsible for what is called "organized preventive population evacuation in case of an ongoing or imminent disaster" in all archeological and other spaces under its responsibility. Clearly, the emergency evacuation of an archeological site of the importance and complexity of the Acropolis of Athens relates to emergency management in the wider area and civil protection procedures in general. To this end, communication and collaboration between different administrative and decision-making

organizations are necessary, and processes for bypassing administrative silos and barriers must be initiated. Collaboration is especially essential among the core involved agents, such as the General Secretariat for Civil Protection, Fire Brigade Headquarters, Hellenic Police Headquarters, Ministry of Health, Ministry of Foreign Affairs, Ministry of Tourism, as well as the Municipality of Athens and Region (Attica).

Furthermore, because of the cultural and economic significance of the Acropolis of Athens, the central government should undertake a leading role and promote effective communication and cooperation with other stakeholders from local administration, the private sector, and civil society. Cooperation among non-government stakeholders could be enhanced through memoranda of cooperation or actions of social corporate responsibility. The key stakeholders to be involved are the Association of Tourist Guides, Association of Employees of the Ministry of Culture, Chamber of Tourism, Maritime Chamber of Greece, and the International Association of Cruise Companies. An organizational unit of the Ministry of Culture could take up advancing risk reduction and emergency management as a horizontal, continuous process.

### 7. Epilogue

To be given the opportunity to examine emergency management in the Acropolis of Athens is both gratifying and intimidating, even more so because, despite the huge number of research projects and studies regarding the Acropolis, there was no previous study focusing on emergency evacuation and the protection of visitors and staff against natural hazards. Finding useful existing material on the topic of our study was therefore challenging, both for the Ministry and for the study group. It took much willingness to collaborate from both sides to identify useful material from different sources and to make the most of the existing firsthand knowledge of staff and officials. At the same time, this was a process of mutual learning and co-working.

The task in hand proved to be far more demanding than expected, far exceeding engineering and technical dimensions and requiring multidisciplinary collaboration between those involved in spatial planning, disaster risk reduction and management, crowd management, and archeology. A study of meteorological and geodynamic hazards that run in parallel also called for multidisciplinary collaboration with geologists and physical scientists. Different disciplines working together proved to be an arduous endeavor; however, at the end it left all involved with a broader understanding of the problem and enriched our proposed solutions. Nonetheless, the exposed multifaceted nature of a seemingly straightforward undertaking brought up novel short-, medium-, and long-term governance challenges that must be met for the proposed solutions to be implemented in the real world.

**Author Contributions:** Conceptualization, M.D. and K.S.; methodology, M.D. and K.S.; software, I.E.; validation, I.E. and M.K.; formal analysis, M.G. and I.E.; investigation, M.G., I.E. and E.L.; resources, M.G. and I.E.; data curation, M.G. and I.E.; writing—original draft preparation, M.G., M.D., I.E., M.K., E.L. and K.S.; writing—review and editing, M.G., M.D., I.E., M.K., E.L. and K.S.; visualization, M.G., I.E. and K.S.; supervision, K.S.; project administration, K.S.; funding acquisition, K.S. All authors have read and agreed to the published version of the manuscript.

**Funding:** This research was funded by the Greek Ministry of Culture and Sports.

**Institutional Review Board Statement:** Not applicable.

**Informed Consent Statement:** Not applicable.

**Data Availability Statement:** Not applicable.

**Acknowledgments:** We would like to express our gratitude to numerous officials and employees of the Ministry of Culture and Sports for their support and fruitful collaboration, as well as for the material and firsthand knowledge they contributed.

**Conflicts of Interest:** The authors declare no conflict of interest.

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
