# Peer review of "Emergency Management against Natural Hazards in the Acropolis of Athens"

_sustainability, doi:10.3390/su142012999_

Round 1

Reviewer 1 Report

See attachment

Author Response

Thank you so much for your suggestions. Please see the attachment

Reviewer 2 Report

This manuscript presents an interesting multidisciplinary approach to emergency management in the Acropolis of Athens. Physical, spatial, human, social, and institutional dimensions are considered to reduce human risk. The paper is well written, which makes it more readable and increases its impact, and covers a topic that is central in the journal scope. The authors provide a detailed description of the problem, simulation assumptions, and are able to contribute to the understanding of the various parameters that play an important role to the risk reduction and sustainability of the Acropolis of Athens and the safety of staff and visitors. In general, I have only some minor comments presented below:

1)     I believe that the title is a bit misleading, as I thought I would see more for the expected meteorological and geodynamic hazards. I would propose better referring to hazardous events in general.

2)     The authors mention that “«Thunderhead Pathfinder software» (TPS) was selected to study and develop the evacuation plan simulation.” Could you please provide more information about it? Any reference?

3)     Lines 163-164: Please change “in a safe manne,r” to “in a safe manner,”.

4)     Line 187: Please explain “nr” the first time you use it, namely number (nr).

Author Response

Thank you so much for your suggestins. Please see the attachment.

Reviewer 3 Report

This paper entitled “Emergency Management against Geodynamic and Meteorological Risks in the Acropolis of Athens” is an extremely interesting scientific approach. risk reduction and preparedness strategy”. It represents a significant advance in the field of science, a continuation of the efforts made by other researchers in the management of risks and emergencies, so the results obtained are sufficiently relevant, leading the research to some separate conclusions.

However, the literature review section of the introduction is very limited and does not offer a broader perspective than would have been expected for a paper like this. Therefore, I recommend that you complete this information using rich scientific achievements in this field of science.

In conclusion, given the topicality of the research, the methodology used, the logical sequence and the results obtained support the publication of the article entitled "Emergency Management against Geodynamic and Meteorological Risks in the Acropolis of Athens" in its current form, with minor changes imposed by journal requirements.

I congratulate the research team for their concerns and achievements.

Good luck!

Author Response

(The authors gave the same response as above.)

Round 2

Reviewer 1 Report

The authors have substantially improved the paper. I suggest its publication in its current state.